# Long-Term Consumption of Green Tea Can Reduce the Degree of Depression in Postmenopausal Women by Increasing Estradiol

**DOI:** 10.3390/nu15214514

**Published:** 2023-10-25

**Authors:** Zhenyu Wan, Xucong Qin, Yuling Tian, Fangcheng Ouyang, Gaohua Wang, Qirong Wan

**Affiliations:** 1Department of Psychiatry, Renmin Hospital of Wuhan University, Wuhan 430000, China; wanzy9889@163.com (Z.W.); psy_qxc@163.com (X.Q.); 2Yichang City Clinical Research Center for Mental Disorders, Yichang 443000, China; tyling2023@163.com; 3BoDe Psychiatric Hospital, Yichang 443000, China; 15971612345@163.com

**Keywords:** depression, green tea, inflammation, hormone

## Abstract

Postmenopausal women face a higher risk of depression due to a combination of social and physiological factors. As a beverage rich in a variety of bioactive substances, green tea has significant effects on metabolism, inflammation and endocrine, and may reduce the risk of depression, but few studies have looked at the effects of green tea on postmenopausal women. Therefore, we designed this study to investigate the effects of long-term green tea consumption on inflammation, endocrine and depression levels in postmenopausal women. We investigated a tea-producing village and eventually included 386 postmenopausal women, both in the tea drinking and control groups. The results showed that there were significant differences in the degree of insomnia, degree of depression, BMI, SII and estradiol between the two groups. And, green tea consumption may reduce the risk of depression through the mediating pathway of sleep, SII and estradiol. In summary, long-term green tea consumption can reduce the risk of depression in postmenopausal women by reducing inflammation and increasing estradiol. This kind of living habit deserves further promotion.

## 1. Introduction

In recent years, with the rapid development of society, people’s pace of life and pressure are increasing and the incidence of various mental diseases is also rising year by year. Among them, depression is the most common mental illness, and its main characteristics include low mood, loss of interest and decreased motivation, accompanied by physical discomfort, decreased appetite and sleep problems, which not only impair individual function, but also lead to suicide in severe cases. In China, the prevalence of depression among adults is as high as 6.8% [1]. There is a significant difference between men and women in the prevalence of depression; before puberty, boys are more likely to suffer from depression, but after the age of 18, girls are significantly more likely to suffer from depression than men, almost twice as likely. Even in older age groups, the incidence is significantly higher in women than in men [2,3]. The gender differences in depression rates are influenced by a combination of factors, with research suggesting that women have a higher genetic risk and are more susceptible to environmental stress [4,5,6]. Fluctuations in sex hormones are also an important reason for the increased risk of depression in women, especially postmenopausal women who face rapid hormonal changes and are at a higher risk of depression compared to premenopausal and menopausal transition [7,8,9]. Considering the gradual aging of postmenopausal women, as well as the decline in cognitive and physical ability, it can be difficult to bear the impact of depression; so, how to reduce their risk of depression is important. However, due to the sluggish treatment of depression, reducing the risk of depression in postmenopausal women through improving their lifestyle has been widely advocated. Among this, tea drinking as a healthy and easily accessible form of health care has been widely explored and studied.

Green tea has always been an important part of the human diet and is widely consumed due to its low price and easy availability, being the second most popular beverage after water. Furthermore, a growing number of studies have shown that green tea is rich in natural bioactive ingredients that not only help enhance cardiovascular and metabolic health [10,11,12] but also have cancer prevention effects [13,14,15,16,17].

The relationship between green tea consumption and depression has also received extensive attention from researchers. Although there is substantial evidence that green tea consumption has antidepressant effects [18,19,20], the mechanism of action has not been fully elucidated and may work through a combination of ways such as reducing oxidative stress [21,22], being anti-inflammatory [23] and improving hormones and gut microbes [24,25,26]. In addition, since there is no uniform standard for the observational indicators of “tea drinking”, the frequency of tea drinking, the intake of tea and the weight of tea are all studied as observational indicators [27,28], which may lead to the fact that the antidepressant effects of green tea are still controversial and more evidence is still needed [29,30]. As a high-risk group of depression, postmenopausal women face a decline in estrogen and an increase in inflammation levels [31,32], and whether green tea consumption can improve these adverse factors deserves more attention. However, volunteers who drank green tea over a long period of time almost replaced water with tea; hence, their daily consumption was very large, so it is difficult to accurately estimate their daily tea consumption or tea weight. Therefore, in this study, we only focused on the lifestyle habit of “tea drinking” and explored whether the lifestyle habit of long-term consumption of green tea can improve endocrine and inflammation in postmenopausal women and effectively reduce the risk of depression through controlled studies.

## 2. Materials and Methods

### 2.1. Study Design and Sample Recruiting

The study was conducted between June 2023 and August 2023 in a tea-growing village where a large proportion of the population began drinking green tea as adults. We recruited participants through local leaders and conducted questionnaires through trained psychiatrists and nurses. The questionnaire included basic socio-demographic information, green tea consumption, sleep and depression. We defined the tea drinking group as “drinking green tea at least 6 days per week, at least 1 cup (500 mL) per day, for at least 20 years,” while the control group never consumed green tea. People who met the criteria and passed the inclusion and exclusion criteria were asked to sign an informed consent form and then had their blood collected on an empty stomach before 9 a.m. Each person who participated in the survey received a gift. Finally, 386 people met the inclusion criteria and completed the questionnaire and blood collection and were finally included in the study.

Inclusion criteria: (1) voluntarily participating in the study and providing written informed consent; (2) participants were able to understand the meaning of the questions in the questionnaire and complete the questionnaire; (3) cooperating with specimen collection.

Exclusion criteria: (1) a history of mental illness or a family history of mental illness; (2) recent major setbacks; (3) presence of endocrine or organic diseases.

All respondents participated voluntarily under the premise of written informed consent and could quit at any time. All methods during the survey were carried out in accordance with relevant guidelines and regulations. Ethical approval was obtained from the Renmin Hospital of Wuhan University.

### 2.2. Variables

#### 2.2.1. Basic Socio-Demographic Information

Basic information includes age, marital status (unmarried, married, divorced, widowed), education level (illiteracy, primary school, junior high school, above junior high school), annual household income (less than 1369, 1369–4100, 4100–10,950, 10,950 and above, unit in US dollars), religious belief, smoking and alcohol consumption, residence status, and only child (yes or no). The PSSS (Perceptive Social Support Scale) was used to measure the degree of social support. The PSSS has 3 subscales, including 12 items in total, and each item adopts the 1–7 scale scoring method. The higher the score, the higher the social support of an individual. Social support from family, friends, relatives and colleagues can be effectively measured, with good reliability and validity [33]. The green tea drinking situation was conducted through a self-designed questionnaire. The interviewer would ask about the type, frequency, consumption and years of tea drinking of the visitors. People who had consumed green tea at least 6 days a week and at least 1 cup (500 mL) per day for more than 20 years were included in the tea drinking group. In contrast, people who never drank tea, or barely drank tea (<once/half a year) were included in the control group.

#### 2.2.2. Sleep Patterns and Depression Levels

The insomnia severity index scale was used to screen for insomnia and assess its severity. The scale consists of 7 items, each item 0–4 points a total of 28 points; the higher the score represents the more serious insomnia. A score of 0–7 is rated as no insomnia, a score of 8–14 is rated as mild, a score of 15–21 is rated as moderate, and a score of 22 or above is rated as severe insomnia. The validity and reliability of this scale has been proved, and it has been widely used in investigations and research [34,35].

The levels of depression were collected using the PHQ-9 (Patient Health Questionnaire), ranging from 0 (not at all) to 3 (almost every day). The higher the score, the more severe the symptoms of depression. The reliability and validity of the questionnaire has been proved [36,37].

#### 2.2.3. Hormone Levels and Inflammatory Markers

Participants were asked to be fasting, and blood was collected in the morning (7–9 a.m.) for hormone and inflammation analysis, using ELISA to measure estradiol, testosterone, TSH (Thyrotropin), FT3(Free Triiodothyronine), FT4(Thyroxine) and inflammation markers. The SII (Systemic Immune Inflammation Index) was used to measure inflammation. The SII is a novel inflammatory biomarker based on lymphocyte, neutrophils and platelet counts, which can reflect the local immune response and systemic inflammation in the whole body. Currently, the SII has been applied in the prediction and correlation studies of a variety of diseases [38,39,40]. However, it should be emphasized that the SII, as a calculated inflammatory index, cannot accurately reflect the specific changes of lymphocytes, neutrophils and platelets.

### 2.3. Statistical Analyses

Analysis was performed using SPSS Statistics 23.0. Categorical variables were represented by N and percentage, quantitative data were represented by mean and quartile, and differences between groups were measured by Chi-square test and non-parametric test. To test the correlation between green tea consumption and the depression level, univariate analysis takes into account all factors that could influence the depression level, followed by ordered logistic regression analysis. To further explore the indirect effects of tea drinking on depression, path analysis was performed by structural equation model using Amos 26.0. The bootstrapping method (5000 resamples) was employed to estimate the 95% confidence intervals (CI) of the mediators’ indirect effect. When the 95% CI did not contain zero, the indirect effect was considered statistically significant1. The level of significance was set at 0.05, two-tailed [41].

Variables whose sample size did not meet statistical requirements were not included in the analysis or were combined with other groups and then analyzed.

## 3. Results

### 3.1. The Characteristics and Differences of Two Groups on Different Variables

A total of 386 people participated in the study; most of whom consumed green tea (57.3%). There were no significant differences between the two groups in terms of education level (*p* = 0.781), annual income (*p* = 0.740), type of work (*p* = 0.336), being an only child and living alone (*p* = 0.647, *p* = 0.511). In addition, there were no significant differences in sleep duration (*p* = 0.992), systolic blood pressure (*p* = 0.383), diastolic blood pressure (*p* = 0.554), TSH (*p* = 0.792), FT3 (*p* = 0.219), FT4 (*p* = 0.984) and testosterone (*p* = 0.297) between the two groups. However, there were significant differences in the degree of insomnia (*p* = 0.003), degree of depression (*p* = 0.002), BMI (*p* < 0.001), SII (*p* = 0.033) and estradiol (*p* < 0.001) between the two groups. (Details are provided in Table 1 and Table 2).

### 3.2. Relationship between Two Groups and Degree of Depression

The factors that may affect the degree of depression were included in the regression model, and the relationship between green tea consumption and degree of depression was explored by ordered logistic regression. The results show that the green tea group has a lower degree of depression (*p* = 0.038). In addition, the concentration of SII (*p* = 0.003) and estradiol (*p* = 0.029) were also significantly correlated with the degree of depression. (For more details, see Table 3 and Figure 1.)

### 3.3. Path Analysis Model

With tea drinking as the independent variable, the degree of depression as the dependent variable, SII, estradiol and the degree of insomnia as the mediating variables, a structural equation model was constructed (Figure 1). The fitting indexes of the structural equation model were as follows: χ^2^/df = 1.297, GFI = 0.996, RMSEA = 0.028, CFI = 0.990, AGFI = 0.980, NFI = 0.960, TLI = 0.966. The results showed that there are significant indirect effects of SII, estradiol and insomnia on the relationship between tea drinking and the degree of depression (Table 4 and Figure 2). The total indirect effect of tea drinking on the degree of depression was −0.094 (SE = 0.030, 95% CI [−0.160, −0.043], *p* < 0.001), accounting for 45.2% of the total effect. Considering that the direct effect was not significant (−0.088, SE = 0.061, 95% CI [−0.183, 0.009], *p* = 0.083), tea drinking could majorly influence the degree of depression by the mediating variables including SII, estradiol and the degree of insomnia. The mediating effects of SII, estradiol and the degree of insomnia were all significant, were −0.022 (SE = 0.012, 95% CI [−0.055, −0.003], *p* = 0.025), −0.014 (SE = 0.008, 95% CI [−0.032, −0.001], *p* = 0.030) and −0.059 (SE = 0.025, 95% CI [−0.118, −0.018], *p* = 0.004), and accounted for 10.3%, 6.7% and 28.2% of the total effect, respectively. Figure 2 displayed the standardized path coefficients of the mediation model.

## 4. Discussion

The accumulated evidence indicates that a variety of bioactive substances rich in green tea have significant effects on the absorption and secretion of many hormones. For example, green tea consumption has been found to reduce the body’s ability to absorb LT4, and animal studies have shown that its catechins affect thyroid function in rats [42], resulting in decreased serum thyroxine levels and increased TSH levels [43,44]. Drinking green tea also has a significant effect on sex hormones, but the conclusion is still not unanimous. Two earlier studies have shown that green tea intake in pre-and postmenopausal women is associated with lower estrogen concentrations [45,46], and a double-blind randomized controlled trial showed that supplementation with different concentrations of green tea extract did not produce consistent changes in sex hormones [47]; there are also studies showing that green tea consumption can increase estradiol concentrations [48,49]. Similarly, the effect of green tea consumption on testosterone has been debated. Conclusions that green tea increases, decreases, and does not affect testosterone levels have been provided by researchers [50,51,52,53]. The results of this study showed that the estradiol concentration was significantly higher in the tea drinking group, while the concentrations of TSH, FT3, FT4 and testosterone were not significantly different from those in the control group. This is not consistent with the conclusions of some studies, and we think that the length of green tea consumption may have different effects on hormones. For example, in an animal study for up to 6 months, researchers found that feeding green tea to rats helped increase estradiol concentrations [48], but in a 16-week study, the estradiol concentrations of female rats given green tea polyphenols did not differ significantly compared to controls [49]. Feedback mechanisms may partly explain the controversy.

Some researchers pointed out in the paper that catechins (the main active substance of green tea) can inhibit aromatase activity so green tea consumption will lead to a decrease in estradiol synthesis, but due to the existence of a feedback mechanism, long-term inhibition of aromatase activity may lead to increased expression of aromatase, which may instead promote an increase in estradiol synthesis (Figure 3) [54,55]. There may be similar effects on other hormones. This may also be an important reason for the inconsistent conclusions on the correlation between green tea and hormones as many studies have ignored the length of green tea consumption history. But, this view still needs further research and evidence in the future, and the length of different tea drinking history may have different degrees and even different directions on the secretion of hormones.

At the same time, the study also showed that SII in the tea drinking group was significantly lower than that in the control group, which was consistent with the conclusions of previous studies. Green tea can exert its anti-inflammatory effects in a variety of ways, such as being an antioxidant or inhibiting the production of inflammatory factors [56,57,58].

In addition, the tea drinkers had significantly lower BMIs and less severe insomnia. The improvement of green tea consumption on BMI has been widely confirmed [59,60], and many studies have shown that green tea can help prevent obesity and metabolic syndrome [61,62], and the results of this study provide evidence for the beneficial effect of long-term green tea consumption on BMI. Previous evidence has shown that green tea contains caffeine, which can lead to disturbed sleep and reduced sleep duration [63,64]. However, the results of this study show that long-term green tea consumption has no significant effect on sleep duration, but significantly reduces the degree of insomnia. This may be because the caffeine in tea will affect sleep in the short term, especially resulting in difficulty falling asleep, but for people who drink tea for a long time, the long-term intake of other active substances in tea has a significant protective effect on people’s sleep, helping people to relax and sleep more peacefully [65,66].

The study also showed a significantly lower degree of depression in the green tea drinking group, suggesting that green tea consumption may help reduce depression [29,30]. However, the current research on this point is still controversial. In addition to the lack of standardized observational measures, both the concentration of green tea and the way it was brewed also had an impact on the bioactive substance and the result, which may have contributed to the differences in the study results. In addition, according to the above mentioned factors, a different length of green tea drinking history may have different effects on physiological indicators and may also have different effects on the occurrence and development of depression. For example, in a 5-year follow-up study, there was no significant difference in depression levels between the tea drinking group and the control group at baseline, but the risk of depression was significantly reduced over time in the tea drinking group [67]. All the subjects in this study have a long-term history of green tea consumption, which may be an important reason for reducing the risk of depression, but whether this conclusion is consistent for short-term tea drinkers still needs to be further explored.

In addition, the pathway model found that green tea consumption may not directly improve the degree of depression but may indirectly reduce the degree of depression by affecting other factors. A pathway analysis showed that tea consumption reduced depression by improving insomnia, increasing estradiol concentration and decreasing SII, with their effects accounting for 28.2%, 6.7% and 10.3%, respectively. Many past studies have provided evidence for this. There is a significant bidirectional relationship between sleep and depression [68,69], with either insufficient sleep or poor sleep quality significantly increasing the risk of depression. Inflammation is one of the main pathogenesis of depression. Some inflammatory factors and acute proteins in the blood of depressed patients are significantly increased compared with healthy controls. Reducing inflammation can effectively improve depression-like behavior [70,71,72]. The concentration of estradiol is closely related to the risk of depression in women, and the rapid change in estradiol concentration is an important reason for the high risk of depression in postmenopausal women, and the use of estrogen therapy can significantly reduce the degree of depression in women [73,74]. As a drink rich in a variety of active substances, green tea may still have many potential pathways that can reduce the risk of depression and deserve further exploration. At the same time, tea drinking, as a cheap and convenient lifestyle habit, is easy to be widely promoted, which is of great help to reduce the risk of depression in the whole group of postmenopausal women.

## 5. Advantage and Limitation

This may be the first study to look at the effects of lifestyle habits (rather than quantity or frequency) of long-term green tea use on behavioral and physiological markers in postmenopausal women, showing that long-term tea consumption can help improve sleep, increase estradiol and reduce depression levels in postmenopausal women. However, this study lacked research on green tea intake, frequency and concentration and could not provide relevant recommendations for the “optimal consumption” of tea drinkers. The subjects of the study were postmenopausal women, and there was no survey of other groups. In addition, the green tea drinking group has a tea drinking history of at least 20 years, so whether the conclusions are still consistent for the people with a shorter tea drinking years needs further investigation. In the future, more standardized observation criteria are needed to improve the consistency of research conclusions.

## 6. Conclusions

Our study showed that postmenopausal women with long-term green tea consumption had higher estradiol concentrations and lower BMI, SII and insomnia levels. At the same time, they also had lower levels of depression, which may be achieved by green tea consumption by reducing inflammation and increasing estradiol. In conclusion, this lifestyle habit deserves further promotion, which can help improve hormone and inflammation levels in postmenopausal women and reduce the risk of depression. In the future, an exploration of the effects of green tea consumption on men and other age groups is necessary. In addition to hormones and inflammation, the rich active substances in green tea may have a significant effect on both the brain and gut microbiota, which can be used as a potential pathway for green tea to reduce the risk of depression and, thus, deserves further study.

## Figures and Tables

**Figure 1 nutrients-15-04514-f001:**
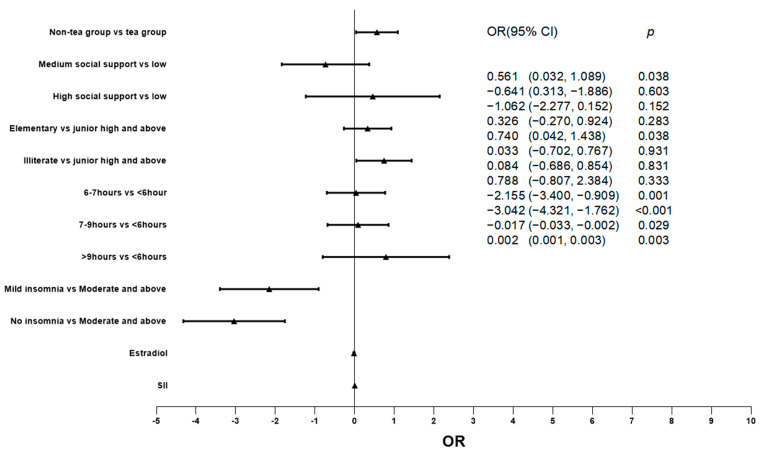
Relationship between different factors and depression degree.

**Figure 2 nutrients-15-04514-f002:**
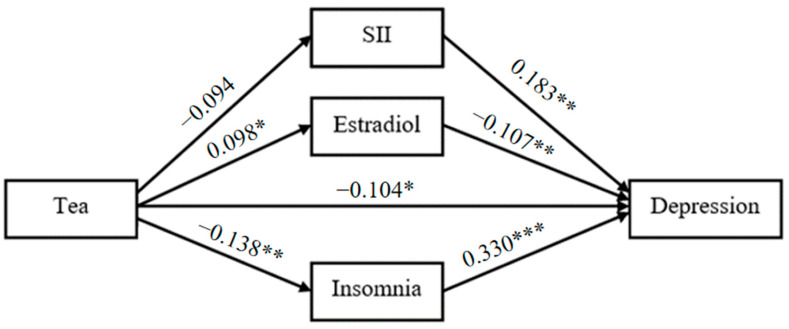
Mediation model. * *p*< 0.05. ** *p*< 0.01. *** *p*< 0.001.

**Figure 3 nutrients-15-04514-f003:**
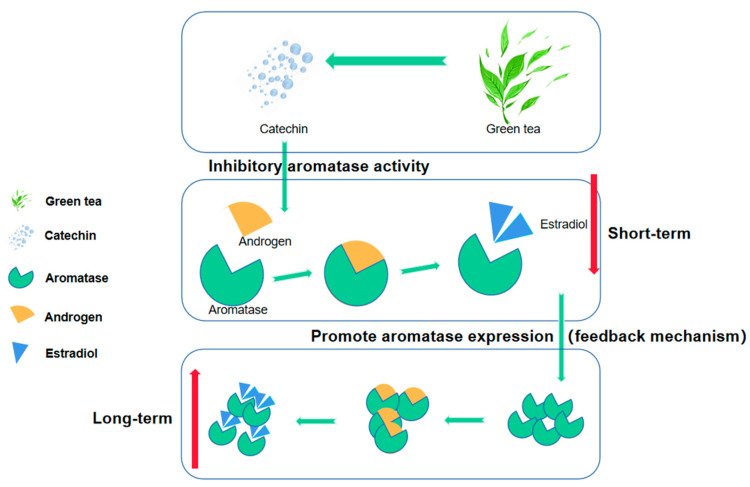
Catechins produced by green tea can affect the conversion of androgen to estradiol by inhibiting the activity of aromatase, which may lead to lower estradiol concentrations in the short term. However, long-term inhibition may activate the negative feedback mechanism, leading to increased expression of aromatase, promoting the conversion of androgen to estradiol and thereby increasing the estradiol concentration.

**Table 1 nutrients-15-04514-t001:** The characteristics and differences between the two groups on different variables.

Variables	Non-Tea Group 165 (42.7%)	Tea Group 221 (57.3%)	*p*
Only child			0.647
Yes	155 (93.9%)	205 (92.8%)	
No	10 (6.1%)	16 (7.2%)	
Educational level			0.781
Illiterate	33 (20.0%)	38 (17.2%)	
elementary	67 (40.6%)	93 (42.1%)	
junior high and above	65 (39.4%)	90 (40.7%	
Annual income ($)			0.740
<1369	79(47.9%)	101 (45.7%)	
1369–4100	61 (37.0%)	90 (40.7%)	
>4100	25(15.2%)	30 (13.6%)	
Live alone			0.511
Yes	11 (6.7%)	11 (5.0%)	
No	154 (93.3%)	210 (95.0%)	
Type of work			0.336
Peasant	117 (70.9%)	171 (77.4%)	
Housewife	34 (18.8%)	34 (14.9%)	
Retired group	14 (8.5%)	16 (7.2%)	
Sleep duration			0.992
>9 h	3.0% (5)	3.2% (7)	
7–9 h	33.9% (56)	34.8% (77)	
6–7 h	30.3% (50)	30.8% (68)	
<6 h	32.7% (54)	31.2% (69)	
Degree of insomnia			0.003
None	119 (72.1%)	189 (85.5%)	
Mild	40 (24.2%)	25 (11.3%)	
Moderate and above	6 (3.6%)	7 (3.2%)	
Degree of social support			0.187
High	94 (57.0%)	146 (66.1%)	
medium	65 (39.4%)	68 (30.8%)	
low	6 (3.6%)	7 (3.2%)	
Degree of depression			0.002
None	113 (68.5%)	185 (83.7%)	
Mild	30 (18.2%)	21 (9.5%)	
Moderate and above	22 (13.3%)	15 (6.8%)	

**Table 2 nutrients-15-04514-t002:** The biological characteristics and differences between the two groups.

Variables	Non-Tea Group 165 (42.7%)	Tea Group 221 (57.3%)	Z	*p*
Age	61.94 (57–67)	61.93 (56–68)	−0.053	0.958
Heart rate	75.12 (69–80)	74.24 (68–81)	−0.730	0.465
Systolic	123.02 (111.0–133.5)	124.83 (109.5–137.0)	−0.873	0.383
Diastolic	77.32 (68.0–84.0)	78.84 (69.0–87.0)	−0.592	0.554
BMI	24.78 (22.89–26.64)	23.23 (21.23–25.10)	−5.140	<0.001
TSH(mIU/L)	3.79 (2.16–3.84)	4.13 (1.69–4.36)	−0.263	0.792
FT4(pmol/L)	17.70 (14.32–18.58)	17.23 (14.58–18.45)	−0.02	0.984
FT3(pmol/L)	4.85 (4.51–5.14)	4.84 (4.57–5.24)	−1.229	0.219
SII	427.02 (281.31–526.07)	387.65 (249.77–485.59)	−2.132	0.033
Estradiol(pg/mL)	27.35 (13.45–30.61)	33.57 (18.63–29.81)	−3.710	<0.001
Testosterone(ng/dL)	15.47 (7.95–15.30)	16.85 (8.28–15.81)	−1.044	0.297

BMI: Body Mass Index.

**Table 3 nutrients-15-04514-t003:** Correlation matrix of model variables.

	DOD	Age	EL	AI	HR	SBP	DBP	TSH	FT4	FT3	BMI	SII	E2	T	DOI	DOSS	SD
DOD	1.000																
age	−0.006	1.000															
EL	−0.127 *	−0.256 ***	1.000														
AI	−0.064	−0.225 ***	0.186 ***	1.000													
HR	0.060	−0.163 **	0.038	0.045	1.000												
SBP	−0.055	0.026	0.062	0.047	0.259 ***	1.000											
DBP	−0.042	−0.119 *	0.089	0.053	0.287 **	0.658 **	1.000										
TSH	−0.048	−0.053	0.050	0.037	−0.071	0.012	−0.027	1.000									
FT4	0.022	0.082	0.005	−0.025	0.153 **	0.099	0.083	−0.238 ***	1.000								
FT3	0.071	−0.070	0.032	−0.054	0.020	−0.023	0.057	−0.077	0.205 ***	1.000							
BMI	−0.000	−0.025	0.092	0.111 *	0.071	0.112 *	0.076	0.032	−0.041	0.053	1.000						
SII	0.149 **	0.026	−0.006	0.011	0.053	0.084	0.078	−0.064	0.053	−0.084	0.041	1.000					
E2	−0.324 ***	−0.095	0.005	−0.032	−0.078	−0.092	0.000	0.032	−0.001	−0.009	0.009	−0.050	1.000				
T	−0.087	−0.070	0.025	0.037	0.046	0.000	0.026	0.066	−0.033	0.005	0.021	−0.059	0.161 **	1.000			
DOI	0.294 ***	0.073 **	−0.089	−0.086	0.003	−0.031	0.019	−0.067	0.016	0.022	0.070	0.053	−0.155 *	−0.143 **	1.000		
DOSS	0.159 **	0.023	0.023	−0.025	−0.013	−0.063	−0.036	0.010	0.006	0.035	0.024	0.009	0.009	−0.002 **	0.132 **	1.000	
SD	0.129 *	0.157 **	−0.087	−0.097	−0.037	−0.003	−0.020	−0.043	−0.009	−0.59	0.010	0.049	−0.116 *	−0.034	0.502 **	0.011	1.000

Abbreviations: DOD: Degree of depression. EL: Educational level. AI: Annual income. HR = heart rate. E2 = Estradiol. T = Testosterone. SBP = Systolic blood pressure. DBP = Diastolic blood pressure. DOI: Degree of insomnia. DOSS: Degree of social support. SD: Sleep duration. * *p* < 0.05. ** *p* < 0.01. *** *p* < 0.001.

**Table 4 nutrients-15-04514-t004:** Mediation analysis results for the relationship between tea and depression.

Mediating Effect Path	Effect Value	Boot SE	Boot LLCI	Boot ULCI	Relative Mediation Effect
Total indirect effect	−0.094 *	0.030	−0.160	−0.043	45.2%
Tea→SII→Depression	−0.022 *	0.012	−0.055	−0.003	10.3%
Tea→Estradiol→Depression	−0.014 *	0.008	−0.032	−0.001	6.7%
Tea→Insomnia→Depression	−0.059 *	0.025	−0.118	−0.018	28.2%

SE, standard errors; LLCI, lower 95% level confidence interval; ULCI, upper 95% level confidence interval; SII, systemic immune inflammation. * indicates the indirect effect is significant (*p* < 0.05).

## Data Availability

The original data will be provided as Appendix A along with the article.

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
