# Peer review of "Long-Term Consumption of Green Tea Can Reduce the Degree of Depression in Postmenopausal Women by Increasing Estradiol"

_nutrients, 2023, doi:10.3390/nu15214514_

Round 1

Reviewer 1 Report

The paper entitled " Long-term consumption of green tea can reduce the degree of depression in postmenopausal women by increasing estradiol" deals with a very interesting and important ersearch question. The effect of green tea consumption on the health of regular consumers is an important issue, and its effect on the health status of postmenopausal wonmen siuffering from depression is of specific interest.

The methodology used here is suitable to answer the most important questions. The results are of great importance. However, the language should be improved (eg fermentation situation, exclusion criteria will be asked, you are the only child, visitors (maybe participants??), retire).

The inclusion criteria starts with a repetition.

The incomes are expressed as USD?

It is not clear why the author mentioned tha blood was collected "each morning". Did I misunderstod that blood was drawn only once from each participant? Please, specify!

Please explain what are the differences between table 1 and 2.

Please discuss the llimitations of the study.

The supporting material contains only Chinese characters. It would be essential to know what is this about if the authors woule like to pubish in an English journal

I do not understand what the "rapid development of society" means. I am not a sociologist (just like the authors) but this term does not sound very scientific (it is rather a polotoical term).

Furthermore, the statement that postmenopausal wmen have "reduced memory" since they are older, is far from a scientific approach

Author Response

Dear reviewer

Thank you very much for taking the time to review this manuscript. Please refer to the attachment for specific modification.

(Due to limitations of the submission system, supplementary files will be submitted to the editor. We feel sorry for the inconvenience brought to the reviewer)

For research article

Response to Reviewer X Comments

1. Summary

Dear reviewer:

Thank you very much for taking the time to review this manuscript, I will answer each of your questions, please find the detailed responses below . And you can see the corresponding revisions highlighted in the resubmitted document.

2. Point-by-point response to Comments and Suggestions for Authors

Comments 1: The methodology used here is suitable to answer the most important questions. The results are of great importance. However, the language should be improved (eg fermentation situation, exclusion criteria will be asked, you are the only child, visitors (maybe participants??), retire)

Response 1: Thank you for pointing this out. We agree with this comment and, as a result, we have corrected the language problems you pointed out.In addition, the paragraph about the introduction of tea was deleted at the suggestion of other reviewers.

Comments 2: The inclusion criteria starts with a repetition.

Response 2: Agree. We have revised.(92)

Comments 3:The incomes are expressed as USD?

Response 3: We gratefully appreciate for your valuable suggestion, and we convert our income into dollars at the current exchange rate.(103-104)

Comments 4:It is not clear why the author mentioned that blood was collected "each morning". Did I misunderstod that blood was drawn only once from each participant? Please, specify!

Response 4: Thank you for pointing out the error. In fact, the volunteers only need to take blood once throughout the study. We have corrected the errors in the article. (128)

Comments 5:Please explain what are the differences between table 1 and 2.

Response 5: Table 1 and Table 2 both reflect people's basic situation, but because of the different data types (Table 1 is classified data and Table 2 is quantitative data), we use two tables for representation.We feel sorry for the inconvenience brought to the reviewer.

Comments 6:Please discuss the limitations of the study.

Response 6: Thank you for your advice, and we have added a discussion of limitations to the text.(310-326)

Comments 7:The supporting material contains only Chinese characters. It would be essential to know what is this about if the authors would like to publish in an English journal

Response 7: Agree,we will submit supplementary materials in English version.

Comments 8:I do not understand what the "rapid development of society" means. I am not a sociologist (just like the authors) but this term does not sound very scientific (it is rather a polotoical term).

Response 8: Thank you for pointing this out.We have made appropriate changes to this paragraph. (28-30)

Comments 9:The statement that postmenopausal women have "reduced memory" since they are older, is far from a scientific approach

Response 9: Thank you for pointing out this error, so we have rephrased it to express the difficulty of accurately estimating daily tea consumption.(68-71)

For review article

Response to Reviewer X Comments

1. Summary

Thank you very much for taking the time to review this manuscript. Please find the detailed responses below and the corresponding revisions/corrections highlighted/in track changes in the re-submitted files. [This is only a recommended summary. Please feel free to adjust it. We do suggest maintaining a neutral tone and thanking the reviewers for their contribution although the comments may be negative or off-target. If you disagree with the reviewer's comments please include any concerns you may have in the letter to the Academic Editor.]

2. Questions for General Evaluation

Reviewer’s Evaluation

Response and Revisions

Is the work a significant contribution to the field?

[Please give your response if necessary. Or you can also give your corresponding response in the point-by-point response letter. The same as below]

Is the work well organized and comprehensively described?

Is the work scientifically sound and not misleading?

Are there appropriate and adequate references to related and previous work?  

Is the English used correct and readable?  

3. Point-by-point response to Comments and Suggestions for Authors

Comments 1: [Paste the full reviewer comment here.]

Response 1: [Type your response here and mark your revisions in red] Thank you for pointing this out. I/We agree with this comment. Therefore, I/we have.[Explain what change you have made. Mention exactly where in the revised manuscript this change can be found – page number, paragraph, and line.]

“[updated text in the manuscript if necessary]”

Comments 2: [Paste the full reviewer comment here.]

Response 2: Agree. I/We have, accordingly, done/revised/changed/modified…..to emphasize this point. Discuss the changes made, providing the necessary explanation/clarification. Mention exactly where in the revised manuscript this change can be found – page number, paragraph, and line.]

“[updated text in the manuscript if necessary]”

4. Response to Comments on the Quality of English Language

Point 1:

Response 1:    (in red)

5. Additional clarifications

[Here, mention any other clarifications you would like to provide to the journal editor/reviewer.]

Reviewer 2 Report

This research is important and brings valuable information for further research and with practical application. The presented research is well-planned, and the manuscript is generally well organized.

Therefore, the paper is of interest, but some points must be reconsidered prior acceptance:

Introduction. Novelty and originality must be clearly stated at the end of this chapter. All these should be highlighted in order to increase the value of the results obtained.

Disccussion. More relevant discussions from the specialized literature would be needed to support that depression can be related to inflammation and low estradiol levels.

The authors could highlight deficient aspects in this review which could be addressed in future research. The authors could specify their future research and development directions in this field. 

In Conclusions is necessary to present the limitations and deficiencies of current studies and perspectives for future studies that could be implemented to cover them.

Author Response

Dear reviewer

Thank you very much for taking the time to review this manuscript. Please refer to the attachment for specific modification.

(Due to limitations of the submission system, supplementary files will be submitted to the editor. We feel sorry for the inconvenience brought to the reviewer)

For research article

Response to Reviewer X Comments

1. Summary

Dear reviewer:

Thank you very much for taking the time to review this manuscript, I will answer each of your questions, please find the detailed responses below . And you can see the corresponding revisions highlighted in the resubmitted document.

2. Point-by-point response to Comments and Suggestions for Authors

Comments 1: Novelty and originality must be clearly stated at the end of this chapter. All these should be highlighted in order to increase the value of the results obtained

Response 1: Thank you so much for your advice, We have highlighted the key points and advantages at the end of the article.(310-315)

Comments 2: Disccussion. More relevant discussions from the specialized literature would be needed to support that depression can be related to inflammation and low estradiol levels.

Response 2: Thank you for pointing this out. We used more literature to support the relationship between depression and inflammation and hormones.(294-304)

Comments 3:The authors could highlight deficient aspects in this review which could be addressed in future research. The authors could specify their future research and development directions in this field. 

Response 3: We gratefully appreciate for your valuable suggestion, and we have supplemented the limitations of the research and the direction of future research.(315-321)

Comments 4:In Conclusions is necessary to present the limitations and deficiencies of current studies and perspectives for future studies that could be implemented to cover them.

Response 4: Thank you for your suggestions, which will make the article more complete. In the conclusion, we summarize the limitations of the study and the prospects for the future.(330-332)

For review article

Response to Reviewer X Comments

1. Summary

Thank you very much for taking the time to review this manuscript. Please find the detailed responses below and the corresponding revisions/corrections highlighted/in track changes in the re-submitted files. [This is only a recommended summary. Please feel free to adjust it. We do suggest maintaining a neutral tone and thanking the reviewers for their contribution although the comments may be negative or off-target. If you disagree with the reviewer's comments please include any concerns you may have in the letter to the Academic Editor.]

2. Questions for General Evaluation

Reviewer’s Evaluation

Response and Revisions

Is the work a significant contribution to the field?

[Please give your response if necessary. Or you can also give your corresponding response in the point-by-point response letter. The same as below]

Is the work well organized and comprehensively described?

Is the work scientifically sound and not misleading?

Are there appropriate and adequate references to related and previous work?  

Is the English used correct and readable?  

3. Point-by-point response to Comments and Suggestions for Authors

Comments 1: [Paste the full reviewer comment here.]

Response 1: [Type your response here and mark your revisions in red] Thank you for pointing this out. I/We agree with this comment. Therefore, I/we have.[Explain what change you have made. Mention exactly where in the revised manuscript this change can be found – page number, paragraph, and line.]

“[updated text in the manuscript if necessary]”

Comments 2: [Paste the full reviewer comment here.]

Response 2: Agree. I/We have, accordingly, done/revised/changed/modified…..to emphasize this point. Discuss the changes made, providing the necessary explanation/clarification. Mention exactly where in the revised manuscript this change can be found – page number, paragraph, and line.]

“[updated text in the manuscript if necessary]”

4. Response to Comments on the Quality of English Language

Point 1:

Response 1:    (in red)

5. Additional clarifications

[Here, mention any other clarifications you would like to provide to the journal editor/reviewer.]

Reviewer 3 Report

Wan et al. investigated here the effects of long-term green tea consumption on inflammation markers (PSSS index), steroid and thyroid hormones, and insomnia/depression scores in postmenopausal women (N=386).

The MS is well-written, objective, and English fluency is adequate. 

From my point of view, the subject is not innovative, despite the authors argue that physiological/molecular mechanisms linking depression and green tea consumption are mandatory to fully understand this interplay. 

The authors presented correlation indexes between all these indexes but, again, the study lacks depth on discuting the possible/reasonable mechanisms involved. Therefore, there are some flaws that need to be corrected in this study before its acceptance in Nutrients/MDPI:

(1) Please, make the Introduction shorter and more straight-to-the-point to address the aims of the study. For example: information presented in lines from 50-56 is unnecessary;

(2) From my understanding, the Inclusion criterium regarding regular green tea consumption is not solid. If it is possible, please present a numeric table with more detailed GT consumption - once, twice, 3x day, or week, etc., or, alternatively, depict in details the volunteers´ regular consumption

(3) Please describe, somewhere in Methods, the limitations of using SII index rather than cytokines measurements to evaluate "inflammation" in the volunteers;

(4) I would like to suggest the authors to include an illustration to better explain the mechanism proposed from lines 241-250 (concerning aromatase activity)

(5) Finally, I suggest the authors to reinforce the correlation between the rich catechin-content (antioxidants and anti-inflammatory) in green tea and oxidative stress within brain segments particularly related with depression.   

English fluency & grammar are fine, but a double-check is always recommended. 

Author Response

Dear reviewer

Thank you very much for taking the time to review this manuscript. Please refer to the attachment for specific modification.

(Due to limitations of the submission system, supplementary files will be submitted to the editor. We feel sorry for the inconvenience brought to the reviewer)

For research article

Response to Reviewer X Comments

1. Summary

Thank you very much for taking the time to review this manuscript, I will answer each of your questions, please find the detailed responses below . And you can see the corresponding revisions highlighted in the resubmitted document.

2. Point-by-point response to Comments and Suggestions for Authors

Comments 1: make the Introduction shorter and more straight-to-the-point to address the aims of the study. For example: information presented in lines from 50-56 is unnecessary.

Response 1: Thank you so much for your advice, we have scaled back the introduction to green tea and made the necessary changes and polish to the citation.(51-56)

Comments 2: If it is possible, please present a numeric table with more detailed GT consumption - once, twice, 3x day, or week, etc., or, alternatively, depict in details the volunteers´ regular consumption

Response 2: Thank you for pointing this out. Volunteers drank green tea at least six days a week to be included in the tea drinking group,so we did not show their specific frequency of tea drinking in the article before, but we will later show this in the supplementary document.

Comments 3:Please describe, somewhere in Methods, the limitations of using SII index rather than cytokines measurements to evaluate "inflammation" in the volunteers.

Response 3: Thank you for your suggestions, which will make the article more complete. We have already added the limitations of using SII as an observational measure.(135-137)

Comments 4:I would like to suggest the authors to include an illustration to better explain the mechanism proposed from lines 241-250 (concerning aromatase activity)

Response 4: Thank you very much for your suggestion, which will make our discussion easier to understand. Images are provided below and in supplementary materials.

Comments 5:Finally, I suggest the authors to reinforce the correlation between the rich catechin-content (antioxidants and anti-inflammatory) in green tea and oxidative stress within brain segments particularly related with depression.   

Response 5: We gratefully appreciate for your valuable suggestion, the effect of catechins on the brain is indeed a worthy direction of research. In the following research work, we will observe the effects of tea drinking on brain inflammation and stress, as well as brain imaging changes.These suggestions will be included in the article as future prospects(321-325)

For review article

Response to Reviewer X Comments

1. Summary

Thank you very much for taking the time to review this manuscript. Please find the detailed responses below and the corresponding revisions/corrections highlighted/in track changes in the re-submitted files. [This is only a recommended summary. Please feel free to adjust it. We do suggest maintaining a neutral tone and thanking the reviewers for their contribution although the comments may be negative or off-target. If you disagree with the reviewer's comments please include any concerns you may have in the letter to the Academic Editor.]

2. Questions for General Evaluation

Reviewer’s Evaluation

Response and Revisions

Is the work a significant contribution to the field?

[Please give your response if necessary. Or you can also give your corresponding response in the point-by-point response letter. The same as below]

Is the work well organized and comprehensively described?

Is the work scientifically sound and not misleading?

Are there appropriate and adequate references to related and previous work?  

Is the English used correct and readable?  

3. Point-by-point response to Comments and Suggestions for Authors

Comments 1: [Paste the full reviewer comment here.]

Response 1: [Type your response here and mark your revisions in red] Thank you for pointing this out. I/We agree with this comment. Therefore, I/we have.[Explain what change you have made. Mention exactly where in the revised manuscript this change can be found – page number, paragraph, and line.]

“[updated text in the manuscript if necessary]”

Comments 2: [Paste the full reviewer comment here.]

Response 2: Agree. I/We have, accordingly, done/revised/changed/modified…..to emphasize this point. Discuss the changes made, providing the necessary explanation/clarification. Mention exactly where in the revised manuscript this change can be found – page number, paragraph, and line.]

“[updated text in the manuscript if necessary]”

4. Response to Comments on the Quality of English Language

Point 1:

Response 1:    (in red)

5. Additional clarifications

[Here, mention any other clarifications you would like to provide to the journal editor/reviewer.]

Round 2

Reviewer 3 Report

I believe the authors satisfactorily replied my questions/comments on the first version of their MS.

Author Response

Dear Reviewer:

On behalf of all authors, thank you for your review and I thank you for your review and contribution to the integrity of the article.

Thank you and best regards

Yours sincerely

Zhenyu Wan